# CDK-Independent and PCNA-Dependent Functions of p21 in DNA Replication

**DOI:** 10.3390/genes11060593

**Published:** 2020-05-28

**Authors:** Sabrina Florencia Mansilla, María Belén De La Vega, Nicolás Luis Calzetta, Sebastián Omar Siri, Vanesa Gottifredi

**Affiliations:** Cell Cycle and Genomic Stability Laboratory, Fundación Instituto Leloir, IIBBA-CONICET, Av. Patricias Argentinas 435, Buenos Aires 1405, Argentina; smansilla@leloir.org.ar (S.F.M.); bdelavega@leloir.org.ar (M.B.D.L.V.); ncalzetta@leloir.org.ar (N.L.C.); ssiri@leloir.org.ar (S.O.S.)

**Keywords:** p21^CDKN1A^, CDK, PCNA, TLS, S phase, DNA replication

## Abstract

p21^Waf/CIP1^ is a small unstructured protein that binds and inactivates cyclin-dependent kinases (CDKs). To this end, p21 levels increase following the activation of the p53 tumor suppressor. CDK inhibition by p21 triggers cell-cycle arrest in the G1 and G2 phases of the cell cycle. In the absence of exogenous insults causing replication stress, only residual p21 levels are prevalent that are insufficient to inhibit CDKs. However, research from different laboratories has demonstrated that these residual p21 levels in the S phase control DNA replication speed and origin firing to preserve genomic stability. Such an S-phase function of p21 depends fully on its ability to displace partners from chromatin-bound proliferating cell nuclear antigen (PCNA). Vice versa, PCNA also regulates p21 by preventing its upregulation in the S phase, even in the context of robust p21 induction by γ irradiation. Such a tight regulation of p21 in the S phase unveils the potential that CDK-independent functions of p21 may have for the improvement of cancer treatments.

## 1. A Broadly Accepted Notion: p21 Controls the Cell Cycle through Cyclin Kinases Inhibition 

Before starting to discuss nuclear functions of cyclin kinase inhibitor p21 (also known as p21^Waf/CIP1^/CDKN1A) we mention that, even though the cytoplasmic functions of p21 were described [1,2,3], these are not the focus of this review. p21 is a 165 amino acid protein characterized by high intrinsic molecular flexibility and nondefined tertiary structure unless when bound to other proteins [4]. At its N-terminus, p21 binds cyclins D/E/A and B, and cyclin-dependent kinases CDK4/6, CDK2, and CDK1 [5,6,7]. In order to achieve CDK inhibition, p21 must simultaneously bind to both cyclin and CDKs [8,9,10]. In cells, serum starvation causes p21 accumulation to such an extent that it titrates and inactivates CDK/cyclin complexes, and allows the accumulation of free p21 in G1 [11]. Endogenous p21 upregulation can also be achieved after the treatment of cells with DNA-damaging agents such as γ irradiation, which causes DNA double-strand breaks (DSBs) and other types of DNA lesions, such as DNA–protein cross-links, oxidized bases, and abasic sites. DNA lesions block replication forks, causing the formation of replication intermediates that activate the DNA damage response (DDR). Kinases within the DDR network phosphorylate p53 and its E3 ubiquitin ligase mouse double minute 2 (Mdm-2), impairing their interaction [12]. As a consequence, p53 levels accumulate, and its transcriptional targets are augmented. Because of the two strong p53-responsive elements within the p21 promoter, p21 mRNA levels increase steeply and rapidly upon p53 upregulation [13]. High p21 levels trigger arrest in Phases G1 and G2 of the cell cycle [5,14,15]. In fact, live tracking of single cells expressing p21-green fluorescent protein (GFP) revealed that G1 cells express the highest and most variable p21-GFP levels that correlate strongly with the length of the G1 phase. The levels of p21-GFP in Phase G2 are lower and less variable than in Phase G1, and they correlate to a lesser extent with G2 phase length [16]. By monitoring CDK2 activity in living cells, previous work also showed that p21 levels at the end of mitosis control cell fate, either by allowing CDK2 activity to build up the next cell cycle, or by inhibiting CDK2 and getting the cell into a G0 state [17]. It is, therefore, expected that p21 expression causes a reduction in the number of cells transiting the S phase. Such a change in cell-cycle distribution generates a time window to repair damaged DNA [12]. p21 is also essential for the prevention of endoreduplication and induction of senescence after extended cell-cycle arrest [18,19]. Because of such a central role in cell-cycle control, p21 is accepted as a bona fide tumor suppressor. Such a notion spread widely in the scientific community and is even found in biology student textbooks. While the validity of such conclusions is not in question, this review focuses on evidence indicating that the nuclear functions of p21 are not at all restricted to the inhibition of CDKs. In fact, we extensively discuss functions of p21 in the S phase that are insufficient to inhibit CDKs.

## 2. Can Cell-Cycle Arrest Be Achieved by p21–PCNA Interaction?

With its C terminus, p21 interacts with the proliferating cell nuclear antigen (PCNA), a protein that was discovered in 1978 and was initially described as an essential factor for proliferation [20]. Many years later, in 1996, the first crystal structure of human PCNA was reported, and it happened to be crystalized with a 22 amino acid peptide corresponding to the C terminus of p21 [21]. PCNA is a torus-shaped homotrimeric ring that encircles DNA and slides in the direction of the replication fork. It provides an anchor point for DNA polymerases in the proximity to DNA, favoring their processivity. While PCNA was initially identified as a processivity factor for DNA polymerases [22], it is now clear that it interacts with hundreds of proteins with known functions in cell-cycle regulation, DNA replication, and DNA repair processes [23,24,25,26]. The majority of such interactions happen through a conserved motif on the surface of PCNA, the 16 amino acid long interdomain connecting loop (IDCL), which interacts with the PCNA-interacting protein (PIP) box of protein partners [25].

PIP boxes reside in intrinsically disordered regions of proteins. The PIP box consensus sequence was defined as Qxxϕxxψψ, where, by definition, Q is glutamine, x is any amino acid, ϕ is a hydrophobic residue, and ψ an aromatic residue (commonly Phe or Tyr) [27]. The p21 peptide used in the original PCNA crystal structure (139GRKRRQTSMTDFYHSKRRLIFS160) [21] contains p21 PIP box QTSMTDFY (aa 144 to 151) that fulfills all consensus sequence requirements. The PIP box of p21 is actually a PIP degron that targets PCNA-interacting proteins for proteasomal degradation via E3 ubiquitin ligase CRL4^Cdt2^. PIP degrons are formed by a threonine and an aspartic acid (TD) within the PIP box, and a basic residue (R) in the +4 position from the last residue of the PIP box [28]. A very recent report demonstrated that the outstanding affinity of p21 for PCNA depends not only on such a core, but also on sequences that flank such a region (aa 140 to 143 and aa 152 to 164). Without those sequences, the affinity of p21 for PCNA drops 10 times. Moreover, if such flanking regions are replaced by hydrophilic or negatively charged sequences, affinity is reduced by four orders of magnitude [29]. In conclusion, the whole C terminus of p21, from aa 139 to 164, generates a very strong binding site for p21. It is, therefore, not surprising that p21 binds to the IDCL with much higher affinity than any other known PCNA-interacting protein [30]. Experiments performed in vitro demonstrated that, indeed, p21 is potentially capable of preventing PCNA interaction with replication factors, including DNA polymerase δ, which is in charge of the elongation of the lagging strand [31]. A scheme of the PCNA binding site within the p21 polypeptide is shown in Figure 1. However, it is not yet clear whether p21 levels in cells are sufficient to displace replicative DNA polymerases from active replisomes. It is also difficult to speculate on the consequence of such displacement. While initial parallelism with its effect on CDK activity may suggest that the PCNA-mediated shut-down of DNA replication could be advantageous for cells experiencing DNA damage, at second glance, such a conclusion might be flawed. While CDK inhibition prevents the initiation of DNA replication [32], the interaction of p21 with PCNA at replisomes would block the cell cycle after the initialization of the S phase. In such a scenario, strong reduction in the speed of nascent DNA elongation or its total inhibition would exacerbate the replication stress, a scenario that is not advantageous for the DDR [33,34]. Hence, the role of p21–PCNA interaction in the S phase must not necessarily parallel the model described for CDKs, in which the only effect caused by p21 when interacting with its partners is triggering their inhibition.

PCNA is a highly abundant protein, especially in cancer cells [35,36] Therefore, it is not clear whether p21 levels that can titrate CDKs outside the S phase would ever be sufficient to titrate PCNA within the S phase. While the p21/PCNA ratio was reported to never exceed a 1:1 ratio in cells [37,38], a ratio of at least 10:1 is required to block DNA repair by nucleotide excision repair (NER) in vitro [38,39,40,41]. p21 was reported to block replicative DNA synthesis in vitro as well, doing so when reaching a 100:1 ratio [42,43,44]. When assessing the effect of p21 on PCNA-dependent DNA synthesis in cells, several reports showed a limited or null effect of p21–PCNA interaction on DNA replication [38,45,46,47,48,49,50,51]. However, few studies suggested that p21–PCNA interaction in cells might result in the CDK-independent arrest of the cell cycle outside the S phase [52,53]. The mechanism by which p21–PCNA interaction in the S phase ends up triggering arrest outside the S phase has not been elucidated. The effect of p21 on NER in vivo is not conclusive either. Some reports indicate a null effect of p21 on NER [54,55,56,57,58], while others reported that p21 overexpression causes NER inhibition [41,59]. Such differences may depend on the levels of overexpression and the constructs used. One report attempted to limit p21 overexpression levels to the amounts observed after endogenous p21 induction with daunorubicin and actinomycin D. Under such overexpression conditions, cell-cycle arrest was efficiently triggered by p21. However, this effect was fully lost when the CDK binding site (W49 to R, F51 to S, and D52 to A) was disrupted. In contrast, PCNA binding to p21 did not affect cell-cycle distribution, bulk DNA synthesis in the S phase, or even the NER-mediated DNA repair. [51]. Hence, levels of p21 overexpression that equal the maximal levels reached by endogenous p21 may not suffice to inhibit PCNA-mediated DNA synthesis by replicative DNA polymerases. As p21 levels during S phase are much lower than the p21 levels induced by daunorubicin and actinomycin D, inhibition of replicative DNA synthesis by endogenous p21 is highly unlikely. Moreover, as is discussed in the next section, the low levels of p21 in the S phase cannot be raised in any known scenario, as p21 proteolysis of p21 is highly upregulated in the S phase.

## 3. p21 Levels are Differentially Regulated during and by Cell Cycle

It is an acknowledged fact within the research community that the accumulation of p21 triggers cell-cycle arrest outside the S phase. In this section, however, we focus on a not-so-familiar yet remarkable fact about p21. In the S phase, p21 is never upregulated, even when treating cells with DNA-damaging agents that cause p21 accumulation to an extent that is easily detectable through a Western blot. It is well known that p21 levels in cells are controlled by different mechanisms that include but are not exclusively restricted to the activity of the following E3 ligases: SCF^Skp2^, CRL4^Cdt2^, MDM2/X (or HDM2 in humans), CRL2^LRR1^, and APC/C^Cdc20^. For details on such degradation mechanisms, please refer to [51,60]. Moreover, de-ubiquitination also plays a role in the control of p21 levels in living cells [61]. p21 levels increase steeply after challenging cells with DNA-damaging agents, mainly because of the p53-triggered increase in p21 mRNA. Consequentially, Western blot analysis of whole-cell extracts showed increased p21 protein levels after γ irradiation, daunorubicin, and actinomycin D treatment. It is simplistic, however, to assume that p21 is equally induced in all phases of the cell cycle, especially when taking into consideration that the activity of the above-mentioned ligases is controlled in a cell-cycle-dependent manner. The activity of SCF^Skp2^, HDM2, and CRL4^Cdt2^ peaks at the beginning of and during the S phase, and could counteract the p53-mediated accumulation of p21 [62,63,64]. It has long been known that, in cells that are not challenged by exogenous damage (unperturbed cell cycle), p21 levels in the S phase remain at their lowest [11]. Moreover, p21 degradation follows a strict timing, and the acceleration of p21 degradation causes stalled replication forks in mid-S phase and sensitivity to replication arrest [65]. Live-cell imaging recently confirmed that p21-GFP levels are the lowest and even undetectable in the S phase [16]. Strikingly, this S-phase-specific low p21 level remains after γ irradiation. Single-cell-based analysis and centrifugal-elutriation approaches showed that, while γ irradiation causes p53 upregulation in all phases of the cell cycle, p21 is upregulated outside the S phase, but not in cells transiting the S phase [63]. In that report, it was shown that the E3 ligase involved in p21 degradation was Hdm-2, at least in some cell lines used in the study [63]. SCF^Skp2^ did not contribute to such γ-irradiation-associated p21 proteolysis in the S phase [63]. Since SCF^Skp2^ degrades cyclin–CDK-bound p21 [1], it is possible that the main signal forcing p21 degradation in the S phase is associated with an E3 ligase that is active beyond the G1/S boundary. More recent studies confirmed the more marginal effect of SCF^Skp2^, while the contribution of CRL4^Cdt2^ to p21 degradation was central and required to promote both irreversible S-phase entry and to prevent premature S-phase exit upon DNA damage [16]. After γ irradiation, the CRL^Cdt2^-dependent degradation of p21 was triggered by ATM (ataxia telangiectasia mutatated) activation [66].

Would p21–PCNA interaction then be needed for S-phase onset and shut-down? A recent report that monitored single cells for 20 hours, combining noninvasive live-cell imaging of CRISPR/Cas9-endogenous fluorescent p53 and p21 reporters and computational-based analysis of multiple single-cell events, addressed such questions [67]. In agreement with [63], they showed that γ-irradiation induced very reproducible kinetics of p53 induction in all cells, peaking at 4 hours after treatment. In contrast, p21 accumulation was heterogeneous and in part uncoupled from p53 dynamics. In cells that were in Phases G1 and G2 at the time of irradiation, an immediate p21 response followed p53 induction. Cells that were transiting the S phase at the time of irradiation induced a delayed p21 response whose onset occurred when they reached the S-to-G2 transition [67]. Similar results were reported when using other p53-inducing agents, radiomimetic drug neocarzinostatin (NCS) and nutlin-3 [16,68]. While the mechanisms of p53 induction by nutlin-3 and γ irradiation are dissimilar (nutlin-3 disrupts p53/MDM2 interaction [69], while γ irradiation attenuates such interaction via the phosphorylation of MDM-2 and p53 [70]), both treatments converge into the accumulation of a transcriptionally active p53 that can promote p21 accumulation. Moreover, under this circumstance, live-cell imaging revealed that p21 accumulated in G1 was downregulated in the S phase, and then accumulated rapidly again at the S/G2 transition [16]. Such modulation in p21 levels was intimately associated with the interaction of p21 with PCNA, as disruption of such interaction through a mutation of the p21 PIP box caused the accumulation of p21 in γ-irradiated cells transiting the S phase [67]. Both studies concluded that the downregulation of p21 in the S phase was dependent on the CRL4^cdt2^ ligase, which degrades PCNA-bound p21 in the S phase [28]. In contrast, SCF^Skp2^ and Mdm2, the other two ligases that mediate p21 degradation during the late G1 and early S phases, contributed only marginally to the prevention of p21 accumulation in the S phase of γ-irradiated cells [16,67]. Intriguingly, the expression of a p21 mutant with a disrupted PCNA-binding domain (Q144A, M147A, and F150A [71]) caused arrest in G1 after γ-irradiation [67], suggesting that p21–PCNA interaction prevented rather than promoted cell-cycle arrest, as originally postulated based on its ability to displace replicative DNA polymerases from PCNA [43,44]. Moreover, p21–PCNA interaction is needed to prevent chromosome fusions and micronuclei that accumulate in cells expressing the p21 mutant deficient in PCNA binding [67]. We speculate that such observation may suggest that, although p21 proteolysis is induced by γ-irradiation in the S phase, the interaction of endogenous p21 and PCNA may favor genomic stability before CRL4^cdt2^-mediated proteolysis. Alternatively, liberating p21 from PCNA interaction may cause dysregulated interaction between p21 and CDKs that favors genomic instability with an unknown mechanism. In this regard, it should be mentioned that, in a manner that depends on CDK inactivation, p21 was reported to regulate the homologous recombination-mediated repair of double-strand-breaks (DSBs), a DNA lesion that is induced by γ irradiation [72]. A model depicting the main conclusions of this section is shown in Figure 2.

## 4. p21 Exerts Specific Functions at S-Phase-Specific Level, Regulating Genomic Stability through PCNA Interaction 

In the previous section, we discussed different reports demonstrating that, regardless of p53-dependent p21 mRNA induction, p21 levels in the S phase are strongly downregulated. Moreover, γ-irradiation cannot trigger p21 accumulation in cells arrested in the S phase using ribonucleotide reductase inhibition by hydroxyurea [73]. Such observations indicate that such a limitation of p21 levels in the S phase could never allow p21-dependent block replicative DNA polymerases. However, recent reports that are discussed below indicated that subtle alterations in the levels of p21 in the S phase affect the replication rate, perhaps by regulating the interaction of PCNA with other partners. Such alterations in replication rate are followed by an increase in the frequency of genome-destabilizing events, unmasking a crucial function of p21–PCNA interaction during the DDR activation in the S phase.

(a) p21 levels influence the turnover of licensed replication origins. By exploring markers of active replication (KI67) and p21 levels in human tumors, unexpected colocalization of p21 and ki67 was observed in approximately 5% to 7% of cells within p53-deficient tumors [71]. Since such results indicated that the S phase of p53-deficient cells could tolerate high p21 levels, cellular models that recapitulated the results were generated. SAOS-2 cancer cells and Li-Fraumeni-derived fibroblasts were engineered to express inducible p21 to levels that were similar to those induced by daunorubicin in p53-proficient MCF-7 cells [71]. After the initial loss of proliferation capacity, a fraction of p21-overexpressing cells evaded arrest/senescence, displaying upregulation of CDT1, CDC6, and ORC, which are all regulators of replication origin licensing [74]. Because p21 has the strongest PCNA-binding affinity, such a result suggested that excess p21 may cause CLR4^Cdt2^ ligase saturation. The dependency on PCNA was confirmed using a p21 mutant with a disrupted PIP box (Q144A, M147A, and F150A), which did not cause CDT1 and CDC6 upregulation [71]. p21-overexpressing cells acquired the capacity to rereplicate. Moreover, replication-fork progression was asymmetric and slow, and DSBs accumulated in a manner that depended on CDT1 and CDC6 overexpression. Because p21 overexpression caused the downregulation of RAD51 [71], DSBs were processed by error-prone pathways, including RAD52-dependent break-induced replication (BIR) and single-strand annealing (SSA) [75], and possibly nonhomologous end-joining [71]. Because both rereplication and error-prone DSB repair fuel genomic instability, these observations indicate a pro-oncogenic role of p21, at least in p53-negative cancer [71,75]. Reinforcing the link between p21 excess in S phase and the instability of the genome, in *Caenorhabditis elegans*, p21 accumulation triggered by CRL4^Cdt2^ and SCF^Skp2^ inactivation causes nuclear retention of CDC6 and rereplication [76]. We conclude that these reports demonstrate that, by competing with other PCNA partners, p21 modulates PCNA-dependent functions in the S phase.

(b) p21 levels influence replication-fork speed. Endogenous p21 levels in unperturbed p53-expressing cells are lower than in the scenario discussed in (a). Such p21 is considered residual as it does not modulate unperturbed cell-cycle progression [77,78]. However, we showed that such low p21 levels interact with PCNA at DNA replication factories [79]. When downregulating p21, both in primary and tumor cells, nascent DNA elongation is impaired, hence suggesting that p21 is a positive regulator of undamaged-nascent DNA elongation [79]. Such a role of p21 requires an intact PIP box, prevents the accumulation of replication stress markers, and protects genomic stability. p21 downregulation causes the augmentation of common fragile-site instability and the accumulation of micronuclei in a manner that depends on the loss of p21–PCNA interaction [79]. All defects observed when p21 is downregulated were phenocopied in HCT116 cells in which p21 was stably knocked out [14,79]. Mechanistically, such a role of p21 depends on its ability to displace PCNA partners from the replisome. In this case, the PCNA partner that triggers the above-mentioned defects was the dysregulated loading of specialized DNA polymerase κ to chromatin-bound PCNA [79]. Specialized DNA polymerases are normally recruited to the replisome at replication barriers that stall ongoing forks. In that context, specialized DNA polymerases promote DNA replication across barriers during a process called translesion DNA synthesis. Such a canonical role of specialized DNA polymerases was reviewed elsewhere [77,80]. However, in this case, the dysregulated loading of pol κ takes place on undamaged DNA, hence causing its participation on undamaged DNA replication, not on TLS events. We postulate that the replication speed on undamaged DNA is negatively affected when pol κ is in charge of DNA synthesis, as pol κ is less processive than replicative DNA polymerases δ and ε. Remarkably, similar conclusions were reached when exploring DNA replication and genomic-instability parameters in the context of the downregulation of the de-ubiquitinase of PCNA, ubiquitin-specific protease 1 (USP1) [81]. While replicative DNA polymerases dock on PCNA at its IDCL, specialized DNA polymerases such as pol κ do so by binding both the IDCL and the ubiquitin moiety of PCNA [82]. Hence when upregulating PCNA ubiquitination or downregulating competitors for IDCL binging, the choice of DNA polymerases in charge of the replication of undamaged DNA is altered. Intriguingly, a more recently published report indicated that p21 downregulation does not reduce, but instead increases DNA elongation in ongoing forks [83]. Both reports [79,83] used the same cell line, but different siRNAs. Another difference between those findings is that Mansilla et al. indicated no significant increase in replication-fork asymmetry [79], while Maya Mendoza et al. found a certain level of asymmetry [83], suggesting that DNA lesions accumulated in the second experiment settings. No p21 null models were used in [83]. The biochemical mechanism that triggers the upregulation of replication speed in the absence of p21 is yet unknown. Importantly, however, both reports [79,83] revealed that endogenous p21 levels that were considered to be residual can control nascent DNA elongation, being capable of enhancing or reducing replication speed. Intriguingly, endogenous p21 expression in the S phase was recently reported to protect cancer cells from excessive DNA damage in the S phase, and cell killing by Wee1 (MK1775) and Chk1 (AZD6772) kinase inhibitors [84,85]. Such a role of p21 is independent of CDK inactivation, and may also depend on the modulation of PCNA interactions during the S phase [86]. A model depicting the main conclusions of this section is shown in Figure 3.

## 5. Genotoxins Do Not Induce p21 Accumulation in S Phase

The S-phase-specific impairment of p21 upregulation after γ-irradiation, NCS, and nutlin-3 was discussed in previous sections. We also debated evidence indicating that such low p21 levels accumulated in unperturbed or γ-irradiated S phase cells are insufficient to attempt CDK inhibition, but enough to modulate the occupancy of PCNA trimmers at replisomes. p21 is capable of modulating the recruitment of many partners to the PCNA IDCL pocket without interfering with the recruitment of replicative DNA polymerase δ to DNA [51]. Such a result is intriguing because pol δ is also loaded to PCNA through PIP box domains [87]. However, replicative DNA polymerases are multimeric complexes containing PIP boxes in each subunit [87]. Given that PCNA is a trimer, the multiple docking options of replicative DNA polymerases may favor their recruitment to the replisome despite endogenous p21 expression [88]. Consistently, we and others showed that pol δ recruitment to PCNA is unaltered by changes in endogenous p21 expression [47,54,55,88].

Since the levels of p21 in an unperturbed S phase do not suffice to inhibit the DNA synthesis by replicative DNA polymerases, they may not suffice after γ irradiation, NCS, or nutlin-3 treatments as well [16,67,68]. Intriguingly, other genotoxins that do not cause cell-cycle arrest in the G1 and G2 phases of the cell cycle, not only fail to induce p21 in the S phase, but happen to trigger p21 downregulation. Agents such as hydroxyurea, aphidicolin, hypoxia and deferoxamine (DFX, hypoxia mimetic agents) cause p21 downregulation both by negatively regulating its mRNA accumulation and by enhancing its proteolysis [37,73,89,90,91]. Focusing on p21 protein degradation, several reports indicated that proteasome-mediated p21 elimination is enhanced by UV irradiation in a dose- and time-dependent manner [59,78,92,93,94,95]. Other genotoxic agents that can efficiently trigger p21 degradation are methyl methanesulfonate (MMS) [94] and cisplatin [95]. All these agents cause replication-fork stalling followed by ATR activation, a situation that may enhance further signaling events involved in p21 degradation [59,93]. Augmented p21 degradation after DNA damage by UV irradiation depends mostly on CRL4^Cdt2^ and a noncanonical type of ubiquitination at its N-terminus [28,64,76,94,96,97]. Together, these observations highlight a counterintuitive concept: p21 is not upregulated in the S phase, and its downregulation, not its accumulation, is required for optimal DDR after at least some types of genotoxic stress.

## 6. CDK-Independent Effect of p21 Overexpression on Damaged DNA Replication, Cell Death, and Genomic Stability

The results in the previous section indicated that p21–PCNA interaction may be actively prevented after different types of genotoxic stress. A hint on the relevance of p21 downregulation in S-phase post-UV irradiation came from the correlation between the timing of p21 degradation and the recruitment of specialized DNA pol η to replication factories or chromatin-bound PCNA [51]. The recruitment of specialized pol η to DNA is particularly relevant for UV-triggered DNA-damaged response. While replicative DNA polymerases cannot replicate through DNA lesions, such as those caused by UV, specialized DNA polymerases incorporate nucleotides opposite damaged DNA [98]. In mammalian cells, such a role is achieved by pol η, pol ι, pol κ, and Rev 1 in the Y family, and pol ζ in the B family of DNA polymerases. Such polymerases are poorly processive, have reduced fidelity due to its lack of 3’–5’ proofreading activity, and possess active sites capable of accommodating damaged or distorted templates [98]. The DNA damage-tolerance pathway that promotes the exchange between replicative and specialized polymerases at stalled forks is known as translesion DNA synthesis (TLS). The recruitment of specialized DNA polymerases to DNA is favored by the ubiquitination of PCNA, thus providing a docking point for ubiquitin-binding domains in specialized DNA polymerases [24,99]. Despite the average poor fidelity of specialized polymerases, certain TLS events are not mutagenic when using specific DNA lesions as replication templates. For example, pol η-mediated DNA synthesis across the major UV photoproduct, cyclobutane pyrimidine dimers (CPDs), is poorly mutagenic [100,101]. The predisposition to skin cancer that characterizes xeroderma-pigmentosum-variant (XPV) disease, lacking pol η expression, presumably results from a more mutagenic bypass of CPDs by other specialized polymerases [102,103]. Hence, p21 downregulation in the S phase may serve to permit the correct onset of TLS events, perhaps even influencing the TLS–pol timing hierarchy [88]. The negative role of p21 expression for the onset of TLS was demonstrated using a plasmid-based assay that allows the quantification of the efficiency and mutagenicity of TLS events [104]. In such assays, p21 reduces TLS capacity, but increases its accuracy, whereby this role depends on the ability of p21 to bind PCNA [104]. Moreover, PCNA ubiquitination, a post-translational modification of PCNA that promotes the recruitment of specialized polymerases to chromatin-bound PCNA, is also affected by p21 levels. However, both the downregulation and overexpression of p21 were reported to impair PCNA ubiquitination [94,104]. Such contrasting, at first sight, results may indicate a bimodal effect of p21 on PCNA ubiquitination. A similar bimodal effect was previously reported for the p21-mediated modulation of CDK4 activation [105]. The conclusion at that point was that p21 downregulation after UV irradiation may favor TLS activation. If that were the case, then the forced expression of p21 in the S phase should impair TLS, triggering a deficient DDR. To address such a question, p21 stabilization was attempted. The construct used that resists UV-triggered p21 proteolysis contains a 6 myc-N-terminus tag that prevented ubiquitination at its N terminus [106]. A mutation on its CDK-binding domain allows unaltered cell-cycle progression in conditions of p21 overexpression [50,51]. Thus, in a manner that depends fully on PCNA binding, p21 overexpression selectively blocked the recruitment of specialized DNA polymerases η, κ, and ι to replication factories both before and after UV irradiation [50]. Rev1 recruitment to UV-damaged DNA is also fully blocked by the expression of a stable p21 with an intact PIP box [50]. The interaction of specialized DNA polymerases, importantly not the replicative Pol δ, with PCNA in chromatin immunoprecipitations was also prevented by stable p21–PCNA interaction, and was released when the PIP box of p21 was mutated (M147 to A, D149 to A, and F150 to A-from [21]) [51]. The impairment of TLS also affected nascent DNA elongation events, as revealed by the observation of single DNA molecules in DNA spreading assays [50]. Importantly, TLS inhibition by stable p21–PCNA interaction correlates with increased cell death and genomic instability after UV irradiation [50]. TLS inhibition after UV irradiation can be achieved under experiment conditions, not only by p21 stabilization in the S phase. PIP box domains with high PCNA affinity, such as the one of CDT1, also displace Pol η and Pol κ from replication factories, causing cell death [107]. To achieve such an effect, the PIP box of CDT1 requires a disrupting mutation of its degron (a substitution of the R-to-A amino acid in position +4 from the C terminus of the PIP box), which guarantees its stabilization [107]. Such mutation types indeed prevent the degradation of PIP box proteins with high affinity for PCNA [28]. Hence, PIP box domains with high affinity for PCNA can prevent the binding of specialized DNA polymerases to the clamp loader. Conversely, the expression of a chimeric specialized polymerase κ with increased affinity for PCNA also causes a similarly unstable outcome [81]. Hence, a very dynamic equilibrium of p21 interactions is required, ensuring conditions for replicative and specialized DNA polymerases to promote scheduled nascent DNA elongation at replication forks. While on undamaged DNA, p21 displaces specialized DNA polymerases favoring DNA replication by pol δ and pol ε, the encounter of a replication fork with a DNA lesion must promote the rapid CRL4^Cdt2^-dependent proteolysis of p21 to allow the utilization of damaged DNA as templates for DNA synthesis by specialized DNA polymerases [77]. A model depicting the main conclusions of this section is shown in Figure 4. 

## 7. Therapeutic Opportunities for Cancer Treatment

Because TLS events may protect tumor cells by favoring DNA replication, TLS is considered a potential target in cancer therapy [98,108,109]. The protective role of TLS in cancer is revealed by the overexpression of pol η in ovarian-cancer cells, which is evidenced in populations that resist cisplatin treatment [110]. Such an adaptive increase in pol η levels was also reported in fibroblasts treated with UV light [111]. Supporting the relevance of TLS in cancer therapy, the elimination of pol η during UV or cisplatin treatment causes acute replication stress in the S phase, leading to the augmentation of a type of cell death that is not accompanied by micronuclei or chromosome aberrations [112]. Hence, TLS inhibition may cause alterations in the DDR that do not favor tumor adaptation to chemotherapeutic treatments. The effect of cisplatin was enhanced when simultaneously depleting pol η in ovarian-cancer-cell models [110]. The upregulation of miR-93, a microRNA that triggers pol η degradation, also enhanced the effect of cisplatin [110]. The development of drugs interfering with different aspects of the TLS pathway was also attempted with promising results [113,114,115,116]. These compounds, however, cannot achieve complete TLS inhibition. One compound to highlight is T2AA, recently described to inhibit the interaction of specialized DNA polymerases with PCNA [117]. Nevertheless, when used at higher concentrations, it can also inhibit replicative polymerases [117]. Such a result demonstrates that TT2A does not have the specificity for the TLS pathway reported for p21–PCNA interaction [50,51]. Another strategy is to inhibit the mutagenic branch of TLS by disrupting pol ζ and Rev1 protein interactions [118,119]. Recently, a small molecule that disrupts this interaction and sensitizes tumors to cisplatin was developed, also demonstrating in vivo efficacy [120]. One more potentially druggable target in the TLS pathway is PCNA ubiquitination [24,99]. Such a post-transcriptional modification of PCNA can also be inhibited by using AKT inhibitors. Such impairment in PCNA ubiquitination enhances the killing of cancer cells treated with cisplatin, displaying an even more potent effect in homologous recombination-deficient cells [121]. In addition, drugs that target p21 may be of use. The inactivation of cullin E3 ligases with pevonedistat (MLN4924) inhibits the proliferation of melanoma cell lines in vitro through the prevention of p21 degradation [122]. Lastly, nanoparticle-based preclinical studies in which TLS polymerases were targeted with siRNA in combination with cisplatin [123] also indicated that TLS inhibition mediated by PIP box peptides or novel drugs may pave avenues towards the improvement of cancer treatments.

## 8. Concluding Remarks

While the concept of p21 as a cyclin kinase inhibitor is massively accepted by the research community, the possibility that such a renowned function of p21 exerts a “secondary” role is far from being acknowledged. For multiple rounds of cell division, cells that cycle freely may not accumulate enough p21 to inhibit CDKs. However, in every cell cycle, cells express p21 in the S phase in such amounts that are enough to fulfill PCNA-mediated functions. Despite the average Western blot showing a faint, almost undetectable p21 band, such an amount of p21 is enough to regulate key aspects of DNA replication events in every cell cycle. This notion is supported by several reports [16,67,75,79,83,86], which demonstrate that endogenous p21, and specifically the fraction expressed in the S phase, has an important role in the control of unperturbed and damaged DNA replication. Moreover, the upregulation of wild-type [75] and a mutant version of p21 [50,67] impairs replicative DNA synthesis. Intriguingly, the downregulation of p21 also interferes with the optimal processivity of DNA replication [79,83,86]. Hence, any perturbation in the p21 levels in the S phase may have negative effects on DNA replication parameters, ultimately causing the activation of a deficient DDR. As p21 is rarely genetically affected in cancers [1,124], the targeting of p21 may have positive results in the clinic [84,85].

## Figures and Tables

**Figure 1 genes-11-00593-f001:**
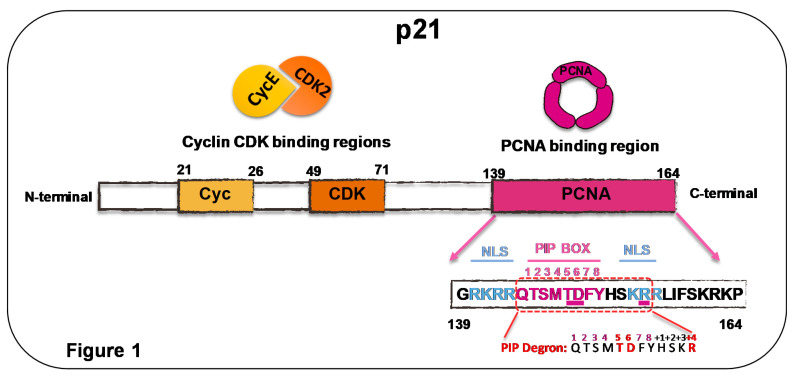
p21 binds cyclin/CDKs and PCNA. Scheme depicting sites of p21 binding partners and detailed regions within PCNA binding region. Threonine (Aspartic acid (TD) at position 148–149 and basic residue (R) in position 155 constitute the degron that is underlined in the model and detailed below. NLS, nuclear localization signal.

**Figure 2 genes-11-00593-f002:**
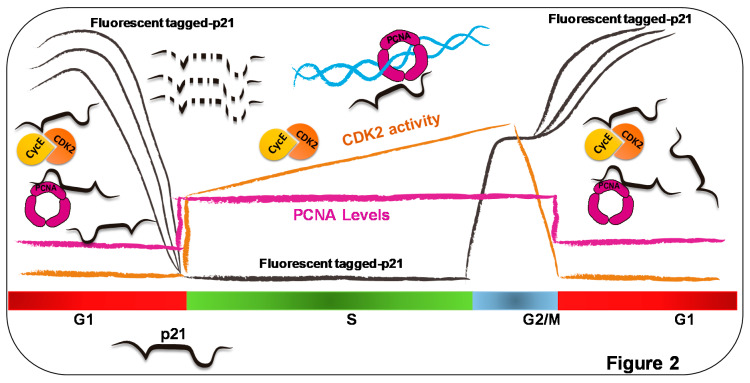
Cell cycle imposes strong regulation on p21 levels in S phase. Levels of fluorescence-tagged p21, PCNA, and CDK activities presented by black, purple, and orange lines, respectively. Endogenous p21 levels not shown, as live image experiments focused on fluorescence-tagged p21. Only in G1 and G2 did p21 levels rise to an extent that could inhibit cyclin–kinase complexes such as CycE/CDK2. Since p21 levels in G1 are very heterogeneous, multiple black lines represent such dispersion in p21 expression. Length of the G1 phase correlates with p21 levels, which were higher than p21 levels in Phase G2 [16]. Whether p21 controls PCNA functions in G1 is still debated, but p21 overexpression levels, which are similar to highest levels reached by endogenous p21 after DNA damage, are not sufficient to modulate replicative DNA synthesis or nucleotide excision repair in G1 [51]. In S phase, levels of fluorescence-tagged p21 were undetectable both in unperturbed conditions and after γ-irradiation, nutlin-3, or neocarzinostatin (NCS) treatment, which are all well-characterized inducers of p21 [16,67,68]. These results indicate that excess of exogenous p21 is strongly prevented in the S phase. Either Cdt2 downregulation or the disruption of the PIP box of p21 allowed p21 upregulation in the S phase, indicating CRL4^Cdt2^ that degrades PCNA-bound p21, tightly controlling p21 levels in the S phase [16,67]. p21 represented as black linear and unstructured peptide; dotted p21 represent p21 degradation.

**Figure 3 genes-11-00593-f003:**
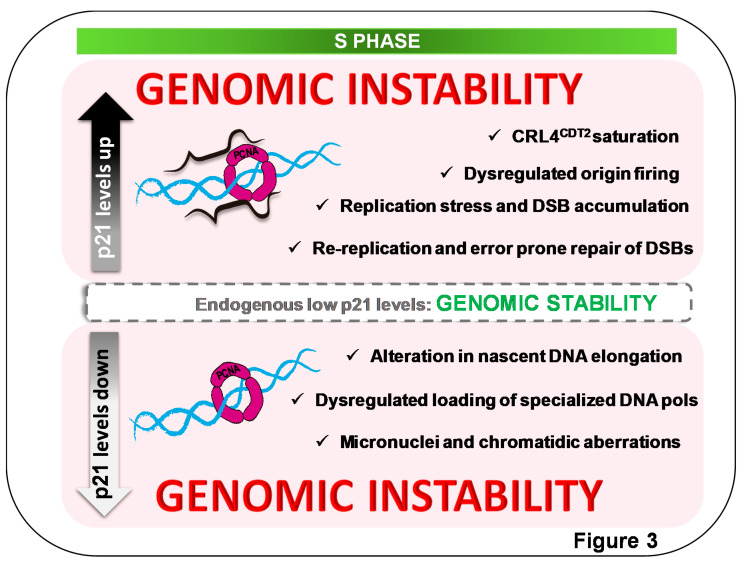
Alterations of endogenous p21 levels in S phase cause genomic instability. Endogenous p21 S normal levels are within the segmented gray box, interacting with PCNA in replication factories [79]. Such p21 levels are required to maintain genomic stability. Both positive and negative changes in p2l levels are detrimental to cell genomic stability. (**bottom**) Downregulation of p21 levels in S causes alterations in nascent DNA elongation speed in a manner that depends on p21–PCNA interaction [79,82]. Mechanism underlying the promotion of nascent DNA elongation by p21 involves the prevention of dysregulated loading of specialized polymerases to sites of DNA synthesis, favoring the utilization of processive polymerases on undamaged templates. Under such experiment settings, p21 downregulation causes genomic instability in a manner that depends on loss of p21–PCNA interaction [79]. The deaccelerating role of p21, oppositely, also depends on p21–PCNA interaction [82]. (**top**) While p53-proficient cells do not allow p21 upregulation in the S phase, p53-deficient cells escaping from a senescent-like phenotype allow DNA replication in the context of p21 overexpression. Increased p21 levels in the S phase saturate the CRL4^Cdt2^ E3 ligase, causing an upregulation of other CRL4^Cdt2^ targets involved in origin firing [71]. Alterations in replication choreography are followed by rereplication events and double-strand-break (DSB) accumulation that are channeled into error-prone repair, further fueling cell genomic instability [71,75].

**Figure 4 genes-11-00593-f004:**
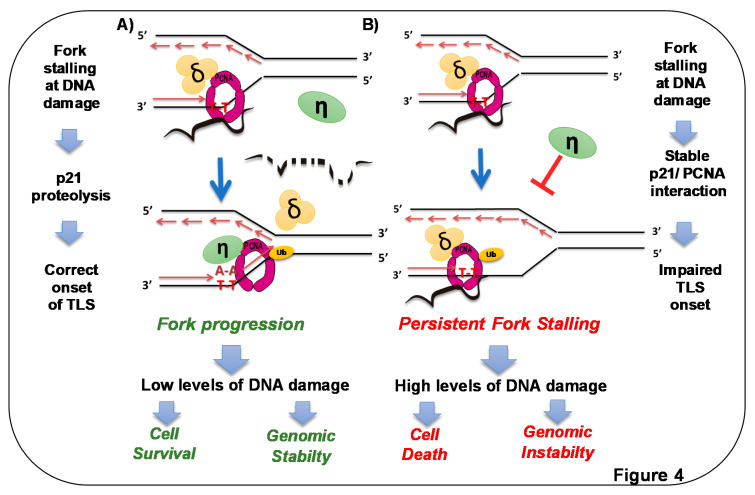
p21 overexpression in S phase triggers cell death by preventing DNA replication across DNA lesions. A) Endogenous p21 is degraded at sites of DNA lesions. Elimination of p21 [50] and concomitant ubiquitination of PCNA [24,98] promote loading of specialized DNA polymerases at forks stalled at DNA lesions [50]. In this way, p21 prevents loading such polymerases in undamaged templates [79], but when DNA lesions are encountered by replication forks, rapid CRL^Cdt2^-mediated removal of p21 facilitates replication across DNA lesions (dotted p21 represents p21 degradation) [50]. B) When p21 degradation is prevented in the context of high PCNA ubiquitination, the recruitment of specialized polymerases to replication factories is impaired. Replicative polymerases stall persistently at DNA lesions, and forks collapse, triggering cell death and genomic instability [50]. Stable p21–PCNA interaction is essential for such a role of p21 in TLS inhibition.

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
