# Peer review of "CDK-Independent and PCNA-Dependent Functions of p21 in DNA Replication"

_genes, 2020, doi:10.3390/genes11060593_

Round 1

Reviewer 1 Report

This review by Mansilla et al. provides a comprehensive overview of p21 functions in DNA replication and repair, focusing specifically on CDK-independent and PCNA-dependent roles for p21 that have not been reviewed extensively elsewhere. Overall, the review covers a wide breadth of material regarding p21 S phase regulation and CDK-independent roles at the replication fork, highlighting the authors’ own work studying the role of low S phase levels of p21 in preventing aberrant TLS polymerase usage. However, to improve the logical flow of this work, the authors should change some aspects of the order in which some material is presented as well as the figure presentation prior to publication.

Major points:

  1. There are some instances in which material is presented in an illogical order. For example, when discussing the effects of p21 levels on fork speed (lines 247-280), the authors cite research on the dysregulated loading of DNA Polymerase k to chromatin-bound PCNA as a potential mechanism. However, the introduction to TLS polymerases does not come until later (line 340). Another example is that the authors begin to discuss studies about the PIP box and PIP degron sequence of p21 prior to talking about the role of these sequences in CRL4Cdt2-mediated targeting of p21 in the following paragraph. A proper buildup to these concepts would be helpful for a more general readership.
  2. To facilitate the flow of the review, it may help to change the ordering of sections. For example, it may help to discuss the mechanisms for the normal cell cycle regulation of p21 (including the role of PCNA as a platform for CRL4Cdt2-mediated proteolysis) prior to talking about the p21-PCNA interaction and cell cycle arrest due to upregulation of p21 in S phase. To better organize topics for discussion, it would also be beneficial to separate the review into two large sections, coinciding with Figure 3: Roles for downregulating p21 in S phase vs. roles for residual p21 in S phase. Subheadings could then follow to discuss the particular studies cited in each section.
  3. The graphical abstract is not very intuitive as drawn. For example, I don’t really understand the highlighting of random words in red for emphasis. Also, the figure legend does not capture the panel on the right. To address scenarios in both panels, perhaps the figure legend should be something like “consequences of deregulation of p21 during the cell cycle”. Then, the left vs. right panels could be separated into two main topics: 1. Aberrant upregulation of p21 outside of S phase. 2. Consequences of downregulation of p21 during S phase.
  4. Figure 2 is also somewhat confusing as drawn. For instance, it took me awhile to realize that the black line was p21 levels, since there is no label next to the line as for CDK2 activity and PCNA levels. Also, the authors should include a separate line for endogenous p21 levels, showing the low-level p21 in S phase, in addition to fluorescent-tagged p21. A major point of this review is that there is some residual p21 that is important for proper regulation of S phase, but that is not emphasized in this figure.
  5. The model in Figure 3 with the arrow for genome instability going in both directions is not intuitive. It would be simpler to separate the figure into two panels listing the consequences of either artificially upregulating (A) or downregulating (B) p21 in S phase. This would convey that there is increased genome instability in both cases.
  6. In Figure 4, the authors should illustrate the ubiquitylation status of PCNA more clearly. A schematic for CRL4Cdt2 targeting would help to show that p21 is being degraded in panel A. Also, an inhibitory sign should be drawn to emphasize that Pol eta recruitment is blocked in panel B due to the persistence of p21 on PCNA.

Minor points:

  1. Although p21-mediated CDK inhibition is not the main focus of this review, an important piece of work that the authors should include in their first paragraph is Spencer et al., Cell, 2013. This paper tracks CDK2 activity using a fluorescent biosensor and relates the level of p21 expression in individual cells to control of the proliferation-quiescence decision following mitotic exit.
  2. Even though the authors explain it in the figure legend, they should also highlight in the figure itself that the “TD” and “B+4” residues constitute the PIP degron of p21.
  3. The depiction of p21 as just a black squiggly line makes it less noticeable in the figures. The authors should write the text “p21” next to the symbol or use a different shape with the text inside it to draw more attention to it.

Reviewer 2 Report

The review article, written by Mansilla et. al. have highlighted the CDK-independent and PCNA-dependent functions of p21 in DNA replication during cell division. This review will be helpful in exploring and understanding the new molecular mechanisms for cell division control rather than only relying on accepted notions. In this article, authors have well-suggested p21 mediated improvement in cancer treatment.

Overall, this manuscript is written well and authors have successfully represented the facts for CDKs-independent and PCNA-dependent function of p21 in DNA replication. Therefore this manuscript may be accepted to publish in the “Genes”.

Author Response

We thank very much reviewers number 2 positive comments about the manuscript.

There were no points to attend in this revision.